# Targeted Anti-Mitochondrial Therapy: The Future of Oncology

**DOI:** 10.3390/genes13101728

**Published:** 2022-09-26

**Authors:** Farzad Taghizadeh-Hesary, Hassan Akbari, Moslem Bahadori, Babak Behnam

**Affiliations:** 1ENT and Head and Neck Research Center and Department, The Five Senses Health Institute, School of Medicine, Iran University of Medical Sciences, Tehran 1445613131, Iran; 2Department of Radiation Oncology, Iran University of Medical Sciences, Tehran 1445613131, Iran; 3Department of Pathology, Shahid Beheshti University of Medical Sciences, Tehran P.O. Box 4739-19395, Iran; 4Traditional Medicine School, Tehran University of Medical Sciences, Tehran P.O. Box 14155-6559, Iran; 5Faculty of Medicine, Tehran University of Medical Sciences, Tehran P.O. Box 14155-6559, Iran; 6Department of Regulatory Affairs, Amarex Clinical Research, Germantown, MD 20874, USA

**Keywords:** ATP, cancer cell, cancer treatment, cancer stem cell, T cell, mitochondria

## Abstract

Like living organisms, cancer cells require energy to survive and interact with their environment. Mitochondria are the main organelles for energy production and cellular metabolism. Recently, investigators demonstrated that cancer cells can hijack mitochondria from immune cells. This behavior sheds light on a pivotal piece in the cancer puzzle, the dependence on the normal cells. This article illustrates the benefits of new functional mitochondria for cancer cells that urge them to hijack mitochondria. It describes how functional mitochondria help cancer cells’ survival in the harsh tumor microenvironment, immune evasion, progression, and treatment resistance. Recent evidence has put forward the pivotal role of mitochondria in the metabolism of cancer stem cells (CSCs), the tumor components responsible for cancer recurrence and metastasis. This theory highlights the mitochondria in cancer biology and explains how targeting mitochondria may improve oncological outcomes.

## 1. Introduction

All living organisms require energy for maintenance, growth, repopulation, and appropriate response to external stimuli. Some organisms are self-sufficient (autotrophs) and acquire energy from sunlight or chemicals. The remaining organisms (heterotrophs) rely on autotrophs to secure energy [1]. Cancer cells are dependent on normal cells for their living and function. Of particular interest, in November 2021, Saha et al. demonstrated that cancer cells can hijack mitochondria (the cell’s energy factories) from immune cells (CD8^+^ T cells and natural killer [NK] cells) via nanoscale tube-like structures [2]. Besides providing energy, mitochondria are essential organelles for cancer cells’ survival and evolution. In addition, mitochondria have a pivotal role in cancer stem cells’ (CSCs) biology, promoting their chemo- and radioresistance [3].

This study aimed to provide a comprehensive overview of mitochondria’s pivotal role in cancer metabolism. The following section explains the mitochondria’s multifaceted role in cancer metabolism and describes how functional mitochondria are vital for cancer survival and progression.

## 2. Mitochondria’s Benefits for Cancer Cells

Mitochondria’s benefits for cancer cells can be classified into four categories, mediating cancer cells’ survival in the tumor microenvironment, immune evasion, progression, and treatment resistance (Figure 1).

### 2.1. Surviving in the Harsh Tumor Microenvironment

Hypoxia threatens human cells by hampering the adenosine triphosphate (ATP) production and excessive reactive oxygen species (ROS) accumulation [4]. Cancer cells can cope with a hypoxic tumor microenvironment (TME) by (1) metabolic switch to glycolysis, (2) enhanced redox homeostasis, (3) protective cell-cycle arrest, (4) pH homeostasis, (5) autophagy, (6) mitochondria hijacking, and (7) promoting angiogenesis [2,5,6].

Accumulating evidence indicates that mitochondria are involved in the strategies mentioned above. This section summarizes the current understanding of the role of mitochondria in tumor hypoxia resistance.

#### 2.1.1. Metabolic Switch to Glycolysis

Cancer cells preserve the ATP/adenosine diphosphate (ADP) ratio in a hypoxic condition by metabolic switch from oxidative phosphorylation (OXPHOS) to anaerobic glycolysis. This phenomenon persists in normoxia, which is known as *aerobic glycolysis* [7]. Hypoxia-inducible factor-1α (HIF-1α) is the master regulator of adaptation to hypoxia. In hypoxia, HIF-1α improves the expression of glycolytic enzymes, including hexokinase 2 (HK2) (the rate-limiting enzyme of glycolysis) and pyruvate kinase M2 (PKM2). In breast cancer cells, HIF-1α promotes glycolysis by upregulating nuclear-factor-erythroid-2-related factor 2 (NRF2) [8]. In addition, HIF-1α prevents pyruvate from entering the tricarboxylic acid (TCA) cycle. This action is mediated by activating pyruvate dehydrogenase kinase 1 (PDK1), which, in turn, impedes pyruvate conversion to acetyl-CoA (the substrate of the TCA cycle) by inhibiting pyruvate dehydrogenase (PDH) [9]. A study on JAK2V617F-positive myeloproliferative neoplasms revealed that HIF-1 signaling is involved in the regulation of genes promoting glycolysis (glucose transporters 1 and 3 (GLUT-1 and -3), phosphofructokinase/fructose-bisphosphatase 3 (PFKFB3), and lactate dehydrogenase A (LDHA)), while shunting pyruvate from TCA cycle (PDK1) [10]. Functional mitochondria enable cancer cells to increase glycolytic flux by stabilizing HIF-1α and facilitating its function [11,12]. The sustained glycolytic pathway provides three benefits for cancer cells: (1) aerobic glycolysis can satisfy the anabolic demands of cancer cells by providing lipids, amino acids, and nucleotides [13]; (2) the pyruvates (interim products of aerobic glycolysis) can serve as an antioxidant and neutralizes the intracellular ROS as a byproduct of cellular metabolism [14]; and (3) normoxic cancer cells can utilize lactate (final products of glycolysis) as an energy source in a process known as *metabolic symbiosis* [13]. It has been evidenced that CSCs have a high glycolysis capacity by expressing high glycolytic enzymes [15]. As a strategy in cancer therapy, targeting glycolytic enzymes can potentially repress stemness properties in CSCs [16,17]. This section denoted and discussed the mechanisms by which functional mitochondria can support HIF-1α stability and function to improve the cancer cells’ capacity to run glycolysis, responding to their high energy demands, proliferation, and surviving in the hypoxic TME. Therefore, it is well interpreted that targeting the cancer cells’ mitochondria can likely reduce their capacity to proliferate and survive in the harsh TME and can make the overall prognosis better.

#### 2.1.2. Redox Homeostasis

Normal cells cannot tolerate hypoxia due to ROS accumulation. The excess intracellular ROS content causes damage to cellular organelles and biomolecules, including DNA, proteins, and lipids. However, cancer cells can tolerate this condition due to an enhanced redox system which is provided by functional mitochondria [18]. Mitochondria are involved in enhanced redox homeostasis of cancer cells in the following ways: (1) Li et al. demonstrated that mitochondria are involved in this process by upregulating antioxidant enzymes (e.g., glutathione reductase, glutathione peroxidase, and glutaredoxins) and redox buffering systems (e.g., glutathione) [18]. (2) The glutathione buffering system requires nicotinamide adenine dinucleotide phosphate (NADPH) to remain reduced. The primary source of NADPH is the pentose phosphate pathway, in which glucose 6-phosphate dehydrogenase (G6PD) is the rate-limiting enzyme [18]. G6PD is directly activated by NRF2, which is upregulated by mitochondrial ROS (mtROS), denoting mitochondria metabolism [8,19]. Moreover, (3) mitochondria can assist in neutralizing the ROS by HIF-dependent glycolytic flux to produce more pyruvates [7]. Experimental evidence demonstrated the antioxidant capacity of pyruvate (see Section 2.1.1) [14,20,21]. A study on chronic lymphocytic leukemia (CLL) cells demonstrated another interaction between intracellular ROS and mitochondria. This study indicated that oxidative stress in CLL cells induced the overexpression of stress-responsive heme-oxygenase-1, which, in turn, promoted mitochondria biogenesis [22]. In other words, oxidative stress, ROS buffering system, and mitochondria form a virtuous circle in CLLs promote cancer pathogenesis in different pathways (see Section 2.3 and Section 3.3). Studies on hepatic CSCs have indicated the high redox capacity of CSCs regulated by CD13 and CD44 [15,23]. It has been demonstrated that combining a CD13 inhibitor with an ROS-inducing chemo/radiation therapy or a CD44 inhibitor with a sulfasalazine can increase intracellular ROS and inhibit tumor progression [24,25]. Therefore, targeting cancer-specific mitochondria can potentially reduce the cancer cells’ capacity to survive in the oxidative TME or to progress (CLL). Moreover, it has been documented that the maintenance of ROS homeostasis determines the maintenance of the CSCs’ phenotype [26]. Hence, tumor recurrence and metastasis rates will likely reduce by targeting the cancer-specific mitochondria.

#### 2.1.3. Protective Cell-Cycle Arrest (Dormancy or Quiescence)

The adaption of cancer cells to survive in harsh TME contributes to tumor recurrence. The dormant state is characterized by mitotic arrest at G_0_/G_1_ phase [27]. In addition, dormancy is a strategy for metastatic cells by which they can remain quiescent in tissues for up to twenty years to find a favorable condition for restarting the cell cycles to proliferate [28]. A study on colon cancer cells indicated that dormancy is through HIF-dependent overexpression of p21 and p27 (two CDK-cyclin inhibitors) [29]. As noted above, HIF-1 requires mitochondria support for proper activity [11,12]. Another study on prostate cancer cells indicated an alternative dormancy signaling pathway, the mitogen-activated protein kinases (MAPK)-dependent p38 activation [30]. It has been evidenced that an increase in mtROS (denoting mitochondria biogenesis) can activate the MAPK-p38 pathway [31]. In leukemic stem cells, quiescence is regulated by the mTOR signaling pathway [32]. Evidence has shown that moderate ROS levels (denoting mitochondrial activity) promote the mTOR pathway [33]. It seems that a competent redox system in cancer cells maintains an ROS balance that is higher than that in normal cells at a moderate level to promote cancer progression (see Section 2.3) [34]. Therefore, mitochondria can participate in the dormancy process via distinct signaling pathways. In support, recent evidence has shown the mitochondria’s reaction to hypoxia. In an in vitro model of dormant breast cancer cells, chronic hypoxia markedly increased mitochondria content and biogenesis [35]. This finding suggests that mitochondria are involved in the regulatory machinery of tumor dormancy. Thereby, inhibiting the cancer cells’ mitochondria would likely increase the rates of proliferative cancer cells, and this, in turn, would improve radiotherapy and chemotherapy efficacy.

#### 2.1.4. pH Homeostasis

Besides hypoxia, acidic pH is another characteristic of TME. This condition is intolerable for normal cells (including immune cells) and leads them to apoptosis. However, cancer cells can tolerate acidic pH by employing a transmembrane glycoprotein called carbonic anhydrase IX (CA IX). It contributes to cancer cells to preserve physiologic pH through bicarbonate influx in cooperation with sodium bicarbonate cotransporters (NBC) and lactate efflux in cooperation with monocarboxylate transporters (MCT). CA IX helps cancer cells to have an increased intracellular pH and a decreased extracellular pH compared with normal cells, both of which are beneficial for cancer cells. The increased intracellular pH facilitates cell proliferation, invasion, and metastasis, while the decreased extracellular pH inhibits immune cells infiltration [36,37]. CA IX is expressed in a wide array of cancer types, including glioblastoma, breast, colorectal, lung, and cervical cancer [38]. It has been demonstrated that CA IX inhibitors (aromatic sulfonamides and disulfonamides) can restrict the cell growth of different cancer cell lines [39]. A study on osteosarcoma cells revealed that mitochondria directly regulate CA IX function [11]. Therefore, targeting cancer cells’ mitochondria can disable cancer cells to provide pH homeostasis surviving in an acidic TME.

#### 2.1.5. Autophagy

In the stressful hypoxic TME, cancer cells preserve cellular homeostasis by degrading and recycling cytoplasmic proteins, lipids, and nonfunctional organelles. A large body of evidence noted that functional mitochondria promote cancer cells to autophagy by increasing the intracellular ROS level, which inactivates the mechanistic target of rapamycin complex 1 (mTORC1) (an autophagy inhibitor) on one hand and activates NRF2 (an autophagy activator) on the other hand [40,41,42,43]. Given the following two well-established assumptions, one might put forward another mechanism by which mitochondria are involved in autophagy: (1) hypoxia-inducing autophagy is mediated by HIF-1α [44], and (2) mitochondria stabilizes HIF-1α and facilitates its function [11]. In breast cancer cells, NRF2 knockdown leads to HIF dysregulation in mediating autophagia [8]. This finding indicates a crosstalk between NRF2 and HIF-1α in regulating autophagy in cancer cells. By inhibiting mTORC1 and activating HIF-1α and NRF2, mitochondria are likely a master regulator of autophagy in cancer cells. Thereby, inhibiting mitochondria would likely impede this rejuvenation strategy of cancer cells and sensitize them to the imposing stresses.

#### 2.1.6. Intercellular Mitochondria Trafficking

A recent study on Lewis lung carcinoma cells revealed that cancer cells generate nanoscale tubes to hijack the CD8^+^ T and NK cells’ mitochondria [2]. This capability enables cancer cells to replace the old, defective mitochondria (degraded by mitophagy) with the new, functional mitochondria from immune cells to reply to the mitochondria demands. We can understand how the existing mitochondria of cancer cells can potentially mediate mitochondria hijacking from normal cells by considering the following assumptions: (1) Upon tunneling nanotube formation, the interaction between mitochondrial Rho GTPase (Miro1) and actins—inside the nanotubes—mediates mitochondria migration from normal cells toward cancer cells; this process is GTP-dependent [2]. (2) It has been indicated that mitochondria’s TCA cycle is the main source of cellular GTP [45]. Targeting mitochondria can likely block the mitochondria’ migration through the nanotubes, thereby shifting the energy balance toward the tumor-fighting immune cells (see Section 3.1), which can improve the response to anti-programmed cell death protein-1 (PD-1) agents [46]. On the other hand, tunneling nanotubes can also mediate mitochondria transfer between cancer cells. In a study on bladder cancer cells, Lu et al. demonstrated that mitochondria transferring through tunneling nanotubes from more invasive (T24) toward less invasive (RT4) cells improved cancer proliferation and invasion [47]. This is strong evidence that functional mitochondria are required for cancer pathogenesis, and cancer cells assist each other to promote their ability to proliferate and invade tissues. In addition, Lu et al. indicated that the formation of mitochondria-transferring nanotubes between bladder cancer cells is regulated by the Akt/mTOR signaling pathway [47]. There is evidence that mtROSs (denoting mitochondrial biogenesis) are an upstream activator of the Akt/mTOR pathway in moderate ROS levels [33]. It has been demonstrated that a competent redox system in cancer cells maintains the ROS balance higher than normal cells at a moderate level to promote cancer progression (see Section 2.3) [34]. Overall, functional mitochondria can also participate in mitochondria transferring between cancer cells as well. Therefore, cancer therapy via targeting mitochondrial would likely interfere with mitochondria trafficking between cancer cells, leading to a reduction in their pathogenesis. Further studies can reveal this notion.

#### 2.1.7. Angiogenesis

In a restrictive TME, cancer cells implicate strategies to find access to oxygen and nutrients supporting their survival and progression. The most established strategy is secreting vascular endothelial growth factor (VEGF), which stimulates angiogenesis to the TME. In a study on lung cancer cells, it has been elucidated that this process is HIF-dependent through direct binding of HIF-1α to the VEGF gene promoter [48]. Besides VEGF, HIF-1α activates the expression of several other angiogenic factors, including placental growth factor, stem cell factor, stromal-derived factor 1, angiopoietin 2, and angiopoietin-like 4 [26]. As noted before, HIF-1α requires mitochondria for proper action [11]. Therefore, one may conclude that anti-mitochondria therapy is more effective than anti-VEGF agents (e.g., bevacizumab) due to inhibiting multiple angiogenic factors. Notably, successfully blocking the angiogenesis can inversely promote cancer proliferation and invasion. This counteracting effect may occur secondary to intra-tumoral hypoxia and increased HIF-1 expression [26]. Therefore, anti-angiogenic agents require complementary agents to impede HIF-1 expression or activity. In a study on EL4 lymphoma cells, a combination of angiostatin and anti-HIF agents was more effective in restricting the lymphoma cells’ growth compared with angiostatin alone [49]. Another potential choice for this aim is anti-mitochondrial therapies that remove the mitochondrial support from the HIF activity (see Section 2.1.1) [11,12].

Collectively, this section demonstrated that functional mitochondria are vital for cancer cells to survive in a harsh TME. Notably, this section also addressed that CSCs have great capacity to run glycolysis and buffering ROSs. It has been evidenced that HIF-1α is the dominant regulator in generation and maintenance of CSCs [26]. A deep dive into the involved molecular mechanisms indicates that HIF-1α upregulates the expression of stem cell factors (e.g., NANOG, OCT4, SOX2, and Krüppel-like factor 4) and TERT gene, an enzyme required for the maintenance of telomeres [26,50]. Targeting cancer-specific mitochondria and removing their support from HIF molecules can help to eliminate CSCs.

### 2.2. Immune Evasion

Functional mitochondria support cancer cells to evade immune surveillance in the following ways:

#### 2.2.1. TME Acidification

Hitherto, it was believed that TME acidification is a cancer cell’s behavior in response to hypoxia. However, a recent study on breast cancer cells revealed that cancer cells can keep acidification even in normoxia. The investigators revealed that low-pH TME modifies the expression of over 3000 genes responsible for tumor invasion, migration, and survival in acidic pH [51]. On the other hand, in low-pH TME, tumor-fighting immune cells lose their function and enter a state of anergy, followed by apoptosis. Cancer cells with functional mitochondria have increased glycolytic flux, which leads to TME acidosis through lactate efflux (the end product of aerobic glycolysis) to the extracellular milieu [9,52]. Furthermore, functional mitochondria can promote TME acidosis by increasing lactate production through HIF-1α-mediated lactate dehydrogenase (LDH) activation and increasing lactate efflux through CA IX mediated MCT activation [9,38]. As noted in Section 2.1.1 and Section 2.1.4, functional mitochondria are essential for proper HIF-1α and CA IX activity [11]. Hence, targeting cancer cells’ mitochondria can disable cancer cells to acidify the TME and helps cancer-specific immune cells recruit the TME, which potentially improves the immunotherapy (ITx) efficacy (see Section 2.4.3). In addition, more acidic TME can improve cancer cells’ progression and resistance. In a study on acute lymphoblastic cells, Bohloli et al. demonstrated that acidic TME helped leukemic cells’ proliferation, invasion, and resistance to apoptosis by doxorubicin (an anthracycline agent) [53]. Therefore, mitochondria-targeting agents can improve the chemotherapy efficacy.

#### 2.2.2. Glucose Influx

In the metabolic competition with immune cells, cancer cells overexpress the glucose transporters (such as GLUT-1) to support their metabolism and make glucose out of the reach of immune cells. Given the importance of glucose for energy production required for proper immune cells function, glucose depletion leads to immune dysfunction [54]. A study on ovarian cancer cells revealed that HIF-1α is the regulating factor of GLUT-1 expression [55]. Moreover, in JAK2V617F-positive myeloproliferative neoplasms, HIF-1α is the primary regulator of GLUT-1 and -3 expression [10]. It has been noted before that mitochondria support HIF-1α expression and function in cancer cells [11]. In a study on thyroid cancer cells, Heydarzadeh et al. demonstrated that GLUT-1 overexpression can also be regulated by the PI3K/Akt pathway [56]. As noted above (see Section 2.1.6), moderate levels of mtROSs (denoting mitochondrial biogenesis) can activate the PI3K/Akt signaling pathway [33]. Therefore, functional mitochondria can increase GLUT-1 expression on cancer cells through HIF-1 and PI3k/Akt pathways. A corollary of this concept is that targeting cancer cells’ mitochondria can reduce glucose uptake by cancer cells, shifting more glucose molecules to the infiltrating immune cells to overcome cancer cells. Moreover, limited available glucose for cancer cells can impede the glycolysis flux, which is essential for their survival, proliferation, and resistance (see Section 2.1.1 and Section 2.1.2).

#### 2.2.3. Mitochondrial Hijacking

Ample evidence has revealed that T cells (as the lead of antitumor immunity) require energy for the proper activation against cancer cells [57]. Mitochondrial hijacking from T cells suppresses immune surveillance by depleting the immune cells’ energy sources. In addition, mitochondrial hijacking from T cells can further block their antitumor function by overexpressing PD-1 molecules on T cells [46]. As noted earlier, mitochondrial trafficking through nanotubes is a GTP-dependent process, and GTP molecules are mainly produced in the mitochondrial Krebs cycle [2,45]. This understanding addresses a crucial concept that blocking mitochondria biogenesis can disable the mitochondria hijacking process. This action can improve antitumor immunity in two ways: (1) by shifting the energy balance toward the immune cells to overcome cancer and (2) by reducing the expression of PD-1 molecules on T cells [46]. Moreover, mitochondria hijacking can reduce the immune cells’ ability to provide prolonged interaction with cancer cells in the context of ITx [46]. Therefore, the improvement of ITx is foreseeable with targeted anti-mitochondria therapy and blocking the hijacking process.

#### 2.2.4. Recruitment of Myeloid-Derived Suppressor Cells (MDSCs) toward TME

MDSCs are one of the principal members of TME. They support tumorigenesis by (1) inhibiting T cells via programmed cell death protein/ligand 1 (PD-L1) expression, uptaking essential amino acids (e.g., cysteine, L-arginine, and tryptophan), and excreting immunosuppressants (e.g., IL-10, TGF-β, nitric oxide); (2) inhibiting NK cells via TGF-β excretion; and (3) inhibiting dendritic cells via IL-10 and nitric oxide excretion. Tumor-infiltrating MDSCs also call regulatory T cells (Tregs) to recruit TME by releasing CC chemokine receptor 5 (CCR5) ligands. Recruited Tregs also have immunoinhibitory effects [58]. Cancer cells lead to MDSCs recruitment into TME by releasing chemokines. A study on hepatocellular carcinoma demonstrated that releasing chemokines by cancer cells is regulated by HIF-1α [59]. As mentioned above, HIF-1α requires mitochondria’s support for the proper action [11]. Therefore, one may conclude that targeting cancer cells’ mitochondria can impede the recruitment of MDSCs into TME, which, in turn, reduces the number of Tregs in TME. In addition, there is evidence that cancer cells support the Tregs’ function and proliferation by providing lactate [60]. Section 2.1.1 and Section 2.1.4 noted that lactate production (by glycolysis) is HIF-dependent and lactate efflux (by MCT molecules) is CA IX-dependent. Given that HIF and CA IX molecules require mitochondria support for the proper action [11], blocking mitochondria can potentially impede the Tregs’ activity and proliferation in TME.

#### 2.2.5. Expression of Immune Checkpoints

Recent evidence has indicated mitochondria participation in expression of PD-L1 on cancer cells. In a study on a melanoma mouse model, the investigators demonstrated that mitochondrial DNA (mtDNA) can be released into the cytosol and triggers PD-L1 expression through the STING–IFN pathway. MtDNA releasing into the cytosol is ATP-dependent, thus elucidating the importance of mitochondria in PD-L1 expression on cancer cells [61]. Moreover, it has been indicated that PD-L1 expression on MDSCs is HIF-dependent [62]. MDSCs’ mitochondria can participate in PD-L1 expression by securing HIF-1α function by producing mtROS [63]. In a colon cancer mouse model, VEGF-A led to PD-1 expression on tumor-infiltrating CD8^+^ T cells [64]. One might link this phenomenon to the cancer cells’ mitochondria; as mentioned above, VEGF-A expression is HIF-dependent, mainly controlled by mitochondria [11,65]. In summary, cancer cells’ mitochondria participate in PD-L1 expression on cancer cells and PD-1 expression on T cells, while MDSCs’ mitochondria participate in PD-L1 expression on MDSCs. This finding indicates that targeting MDSCs’ mitochondria can also help to improve the immune response by reducing PDL1-induced immune inactivation.

#### 2.2.6. Defective Antigen Presentation

One of the main strategies which cancer cells apply to evade the immune system is major histocompatibility class I (MHC-I) downregulation. With this strategy, cancer cells hide their tumor-specific antigens from T cells, preventing adaptive immune response. Defective antigen presentation has been reported in 40–90% of cases of distinct malignancies, including melanoma, colorectal, breast, and cervical cancers, usually associated with poor overall prognosis [66,67]. In a fibrosarcoma mouse model, It has been reported that hypoxia downregulates MHC-1 through an HIF-dependent process [68]. As mentioned earlier, mitochondria are essential for proper HIF-1α action [11]. Other oncogenic pathways can participate in MHC-1 downregulation. It has been demonstrated that the MAPK pathway can decrease the surface expression of MHC-I through STAT1 and IRF downregulation [69,70]. Moreover, there is evidence that an increase in mtROSs (denoting mitochondrial activity) can enhance MAPK activity [31]. These findings note that mitochondria can participate in MHC-1 downregulation through multiple pathways. Hence, targeting cancer cells’ mitochondria removes their support from HIF-1α and MAPK and enhances the antigen presentation to T cells. In this condition, more immune cells infiltrate the TME, and this, in turn, would enhance both intrinsic (without ITx) and ITx-induced immune responses.

#### 2.2.7. Immunosuppressive Mediators

Besides MDSCs and Tregs, cancer cells per se can suppress immune control by releasing immunosuppressants. It has been shown that HIF-1α increases gene expression of IL-10 and TGF-β by direct binding to their promoter [65]. As mentioned, HIF-1α expression and function are dependent on functional mitochondria [11]. However, HIF-1 is not the only regulator of immunosuppressant secretion. It has been evidenced that the PI3K pathway can also mitigate the immune response to cancer by limiting the secretion of proinflammatory cytokines (e.g., IL-12) and increasing the production of anti-inflammatory cytokines (e.g., IL-10) [71]. As noted in Section 2.1.6, a moderate mtROSs level (denoting mitochondrial activity) promotes PI3K signaling pathway [33]. These findings note that mitochondria can mediate the production of immunosuppressants through multiple pathways. Therefore, mitochondria blocking would turn the immunosuppressive TME into proinflammatory gene signature. This change can improve the response to ITx and radiotherapy, both through enhancing effector T-cell infiltration into TME [72,73].

Collectively, this section demonstrated that functional mitochondria are crucial for cancer immune evasion. Targeting cancer-cell- and MDSC-specific mitochondria can improve the antitumor immunity in the context of ITx and intrinsic immunity.

### 2.3. Cancer Progression

Mitochondria generate 90% of the total cellular ROS volume, mainly by complexes I and III of the mitochondrial respiratory chain [74,75]. ROSs are a group of oxygen-containing, highly active, short-lived molecules. ROS in cancer cells is a double-edged sword. On the one hand, it helps cancer progression in moderate levels; on the other hand, it leads to cancer cell apoptosis at high levels [75]. Functional mitochondria give rise to elevated *ROS balance*. It means they elevate and maintain the ROS concentration at moderate levels to help cancer progression but impede damage to the cancer cells’ components [34]. This section explains how mtROSs improve cancer progression through (1) genomic instability, (2) cell-cycle checkpoint evasion, and (3) epithelial-to-mesenchymal transition (EMT), which is a prelude for metastasis.

#### 2.3.1. Genomic Instability

Genomic instability is a hallmark of cancer, and mitochondria can assist genomic instability in several ways. First, elevated mtROS directly damage mitochondrial and nuclear DNA by oxidizing nucleosides [76]. Another mechanism by which mitochondria lead to DNA mutation is by inducing minority mitochondrial outer membrane permeabilization (MOMP). Compared to MOMP (which is the trigger point of apoptosis), minority *MOMP* causes DNA mutation without apoptosis [77]. In esophageal cancer cells, an increase in ROS production and Mcl-1 expression are associated with minority MOMP [78]. Functional mitochondria are involved in minority MOMP through elevating ROS production and securing HIF-1α function, which directly increases Mcl-1 expression [79]. Besides genetic mutations, the inactivation of DNA damage repair pathways is essential for establishing the genomic instability in cancer cells [80]. The direct effect of mitochondria on DNA-damage repair has not been elucidated. Interestingly, one might assume this effect by considering the following two assumptions: (1) HIF-1α leads to downregulation of mismatch repair (MMR) genes [81], and (2) mitochondria secure and conserve HIF-1α function. This paragraph illustrated how functional mitochondria can induce and preserve genetic mutations in cancer cells. With a gradual increase in genomic instability, cancer cells lost their differentiation. This process will be translated to high proliferation and invasion in cancer behavior [82].

#### 2.3.2. Quiescence Evasion

In a growth-permissive TME, cancer cells exit the quiescence state and restart the cell cycle to proliferate. Mitochondria can participate in quiescence evasion in two ways. (1) Extrinsic pathway: β1 integrin is a cell surface receptor that interacts with TME and mediates cancer cells’ invasion and metastasis [83]. In growth permissive TME, β1 integrin activates the FAK–Src–MAPK pathway, prompting cancer cells to restart the cell cycle [84]. An in vivo study on osteosarcoma cells demonstrated that blocking OXPHOS resulted in β1 integrin overexpression [85]. This process is similar to aerobic glycolysis, in which HIF-1α shifts cancer cells’ metabolism from OXPHOS to glycolysis. However, under normoxic conditions, HIF-1 is deactivated [26]. Therefore, one may conclude that quiescence evasion is orchestrated by another transcription factor. Recent evidence on a non-small cell lung cancer (NSCLC) model demonstrated that this process is regulated by Ras, Rap1, PI3K/Akt, and ERK signaling pathway [86]. Moreover, it has been noted that mitochondria are indexed actors in Ras-mediated cancer proliferation. Serasinghe et al. indicated that mitochondria dynamics, shifting from mitochondrial fusion into fission, is required for cancer proliferation [87]. Therefore, determining factors involved in mitochondria dynamics can help to overcome Ras-mediated quiescence evasion. (2) Intrinsic pathway: An elevated ROS level can lead to cell-cycle reactivation. Functional mitochondria can contribute to quiescence evasion by producing more ROSs [88]. Looking back to Section 2.1.3, we may conclude that mitochondria are involved in both cell-cycle arrest and cell-cycle evasion. Further studies are required to reveal underlying mechanistic pathways involved in this dual behavior of mitochondria in different TME conditions. 

#### 2.3.3. Metastasis

EMT is the prerequisite for metastasis of cancer cells by inhibiting cell–cell adhesion and promoting local migration, vascular invasion, and resistance to apoptotic stimuli [89]. EMT and cancer-cell stemness are correlated phenomena regulated by common mediators, including HIFs, SNAIL, and SLUG/SOX9 [90,91]. Interestingly, the p53 tumor-suppressor gene can promote a reverse pathway of mesenchymal to epithelial transition (MET) and differentiation [91,92]. It has been established that ROS promotes EMT through MAPK and PI3K/Akt/mTOR activation, which, in turn, activates downstream SNAIL, matrix metalloproteinase 2 (MMP2), and MMP9 enzymes initiating EMT [75,91]. Moreover, in breast cancer cells, ROSs can lead to EMT through the VEGFA–SOX2–SNAI2 pathway [93]. As noted, functional mitochondria elevate the intracellular ROS balance and maintain it at a moderate level [34]. Another mitochondria-mediated mechanism has been demonstrated in cancer metastasis. In an invasive breast cancer model, the crosslink between β1 integrin and the extracellular matrix was involved in cancer proliferation, invasion, and metastasis [94]. This process is mediated by lysyl oxidase (LOX), which is upregulated by HIF-1α [95]. Mitochondria enhance LOX function by securing HIF-1α function [11]. Therefore, targeted anti-mitochondrial therapy has the potential to disrupt EMT and metastasis. 

Collectively, this section demonstrated how functional mitochondria assist cancer progression. Targeting cancer-specific mitochondria can reduce their ability to de-differentiate, proliferate, and metastasize. Therefore, it can help to improve the treatment results and overall prognosis.

### 2.4. Resistance to Treatment

#### 2.4.1. Chemotherapy

Mitochondria protect cancer cells from chemotherapy in several ways: (1) Most chemotherapeutics trigger cell death through oxidative stress. This is mediated by damage to cancer-cell components and promoting apoptosis [96]. As noted in Section 2.1.2, mitochondria are involved in enhanced redox homeostasis of cancer cells by direct expression of antioxidant enzymes and glutathione, providing nicotinamide adenine dinucleotide phosphate (NADPH) to preserve glutathione at a reduced state and increasing pyruvate production through glycolysis flux [8,11,18,19]. (2) Multidrug resistance (MDR) is mainly due to ATP-dependent multidrug efflux pumps that pump out chemotherapy agents. In a small cell lung cancer model, MDR efflux pumps were upregulated through the NRF2 pathway [97]. As noted before (Section 2.1.2), functional mitochondria stimulate NRF2 function by increasing mtROS [42]; (3) functional mitochondria assist MDR by providing sufficient ATP for ATP-dependent efflux pumps [98]; (4) in breast cancer cells, mitochondria increased doxorubicin resistance by inducing cell cycle arrest [99]; and, as alluded to above (Section 2.2.1), (5) mitochondria can help to create doxorubicin-resistance by mediating TME acidification [53]. Therefore, functional mitochondria enhance the cancer cells’ chemoresistance through (at least) five abovementioned mechanisms. Future clinical trials can examine anti-mitochondrial therapies to improve chemotherapy efficacy.

#### 2.4.2. Radiotherapy

Ionizing radiation can damage cancer cells by direct damage to DNA or dominantly through ROS generation and indirect damages to cellular and mitochondrial components [100]. As mentioned in Section 2.1.2, mitochondria protect cancer cells from radiotherapy by scavenging the generated ROS [8,11,18,19]. As noted in Section 2.3, cancer cells’ mitochondria keep the ROS balance higher than normal cells at a moderate level [34]. An Elevated ROS level activates a focal adhesion kinase (FAK) transcription factor [101]. A recent study on a pancreatic cancer mouse model demonstrated that FAK inhibitors can improve the radiosensitivity of tumors through enhanced CD8^+^ T-cell infiltration [72]. In other words, the FAK signaling pathway induces immunosuppressive TME in pancreatic cancer. A corollary of these concepts is that anti-mitochondrial therapy can serve as a radiosensitizer by reducing the ROS-clearing capacity of cancer cells and improving effector T-cell infiltration. The clinical implication of this concept is to reduce the prescribed radiation doses, thus reducing the treatment toxicities and improving the quality of life. Future preclinical and clinical studies can reveal the radiosensitizing efficacy of anti-mitochondrial therapies.

#### 2.4.3. Immunotherapy

In addition to radiotherapy and chemotherapy, mitochondria can enhance the resistance to ITx. This notion was demonstrated in an in vivo experiment in which blocking the mitochondria trafficking from T cells to cancer cells improved the efficacy of anti-PD-1 ITx [2]. As noted in Section 2.2.3, functional mitochondria can potentially take part in the mitochondrial hijacking process by providing sufficient GTP for Miro1 [45]. This process depletes T cells’ energy and impedes long-term immune surveillance [46]. This defensive mechanism of cancer can also involve other modes of ITx, including adoptive cell therapy and cancer vaccines. Therefore, targeted anti-mitochondrial therapy can improve the immune cells’ energy content to better recruit the TME and counteract the cancer cells. This strategy can act as an adjuvant to diverse ITx approaches.

Collectively, this section demonstrated how anti-mitochondrial therapies can potentially tackle cancer-treatment resistance.

## 3. Discussion

### 3.1. An Energy Battle between Immune and Cancer Cells

This article demonstrated the crucial role of mitochondria in cancer cells’ survival, progression, and confrontation with immune cells. In the struggle between immune and cancer cells, each party with a higher energy level can win the battle. More functional mitochondria empower the cancer cells and enable them to overcome their opponent, the immune cells. As alluded to above, mitochondrial hijacking from immune cells upgrades the cancer cells’ resistance to anti-PD-1 antibodies [2]. This finding supports the hypothesis that T cells’ mitochondria content determines response to anti-PD-1 ITx. In January 2021, Akbari and Taghizadeh-Hesary et al. described how T cells’ mitochondrial activation can improve the response to anti-PD-1 antibodies by improving recognition (through PD-1 downregulation on T cells) and providing energy for long-term T-cell activation [46]. This strategic finding can introduce a new potential theory in oncology, *the energy battle*. In this theory, shifting the energy balance toward the immune cells can improve clinical outcomes. Theoretically, leveling up the immune cells (against cancer cells) can potentially serve as monotherapy. Immune cells with stronger mitochondria are more efficient in all phases of cancer-cell recognition (through PD-1 downregulation), activation, proliferation, migration, and cancer-cell killing [46,102,103]. All of these phases are ATP-dependent [46,57]. On the other hand, cancer cells with weaker mitochondria cannot tolerate the bulk of ROSs generated in the hypoxic TME and procced to apoptosis. Shifting the energy balance toward the immune cells is accessible by improving T cells’ mitochondria in quantity and quality. For the former, the T cells’ mitochondria numbers can be saved by blocking mitochondrial hijacking [2]. The mitochondria quality can increase by two strategies: (1) improving the lifestyle with regular exercise [104], a low-SDA (specific dynamic action) diet [105], good sleep [106], a healthy weight [107], and smoking cessation [108]; and (2) mitochondria boosting agents (e.g., activators of adenosine monophosphate-activated protein kinase (AMPK), mammalian target of rapamycin (mTOR), and peroxisome proliferator-activated receptor-gamma coactivator 1-alpha (PGC-1α)) [109]. This theory might improve the ITx efficacy in future studies.

### 3.2. Mitochondria Improve Treatment Resistance

Despite considerable advances in cancer treatment, cancer recurrence is frequently seen. It has been reported in 50% of patients with soft-tissue sarcoma, 85% of patients with ovarian cancer, and almost all patients with glioblastoma [110]. Cancer cells can develop resistance to the available treatments through specific genetic and epigenetic changes. For instance, resistance to radiotherapy by amplifying ROS clearing system, resistance to chemotherapy by MDR efflux pumps, cell-cycle arrest, TME acidification, ROS clearing, and resistance to ITx by depleting T cells’ mitochondrial content through mitochondrial hijacking. This article demonstrated that mitochondria are common actors in these resistance mechanisms. Moreover, in response to targeted therapies, cancer cells can circumvent the blocked pathway through many different mechanisms [111], including (1) restoration of the targeted molecules (e.g., BCR-ABL kinase reactivation in imatinib therapy of chronic myelogenous leukemia) [112], (2) activation of upstream and downstream signaling proteins (e.g., MAP kinase signaling restoration in vemurafenib therapy of melanoma) [113], (3) histologic transformation (e.g., transformation into small cell carcinoma in tyrosine kinase therapy of EGFR mutant NSCLC) [114], and (4) adaptive signaling to promote survival (e.g., HIF-dependent cell-cycle arrest in doxorubicin therapy of breast cancer) [99]. The current literature indicates that targeting cancer through different mechanisms can improve clinical outcomes. To better delineate this notion, the following example is presented. Over the last two decades, the six-month PFS (progression-free survival) of patients with platinum-resistant ovarian cancer has improved from 30% in chemotherapy-alone [115] to 47% in the chemotherapy plus anti-VEGF [116] to 53% in the chemotherapy plus anti-VEGF plus anti-PD-1 [117]. This improvement in oncological outcome is at the expense of more toxicities. This article indicated that anti-mitochondrial therapy can block/attenuate multiple resistant mechanisms to improve the treatment responses for the available anticancer approaches, including chemotherapy, radiotherapy, ITx, and targeted therapies.

### 3.3. Reactive Oxygen Species: The Main Weapon of Mitochondria

This paragraph describes how mitochondria-derived ROSs can serve as signaling molecules in cancer cells. “Reactive oxygen species” is an umbrella term denoting highly reactive molecules formed from oxygen (O_2_). The common forms of ROS are hydrogen peroxide (H_2_O_2_), superoxide (O_2_^−^), and hydroxyl radical (•OH). ROSs are mainly generated during mitochondrial oxidative metabolism in cancer cells and cancer-associated fibroblasts [118]. Evolving evidence has explored the role of mtROSs as signaling molecules in cancer pathogenesis. Hydrogen peroxide molecules can promote different oncogenic signaling pathways, including HIF-1α, PI3k/Akt, and MAPK [119]. As noted in the previous section, these pathways are the main regulators of cancer metabolism in all disciplines of survival in TME, immune evasion, progression, and treatment resistance. Superoxide molecules also have signaling effects in cancer cells. It improves the stabilization of HIF-1α and upregulates adenosine monophosphate–activated protein kinase (AMPK), as the main regulator of cancer cells’ growth and proliferation [120]. Given the diverse and crucial effects of mtROS as signaling molecules, anti-mitochondrial therapies serve as a potential anticancer treatment, influencing cancer survival, progression, and resistance.

### 3.4. Cancer Stem Cells Can Be Defeated by Targeting Mitochondria

Cancer stem cells are responsible for cancer initiation, progression, resistance, recurrence, and metastasis. It has been evidenced that CSCs activate mitochondrial stress pathways in response to stressors such as radiation, chemotherapy, or hypoxia. This contribution is multidimensional by regulating stemness, quiescence, and treatment resistance [121]. Recent studies on glioblastoma stem cells (GSCs) demonstrated the pivotal role of mitochondria in GSCs biology. In this study, Sighel et al. realized that the quinupristin/dalfopristin (Q/D) combination suppresses GSCs’ growth by inhibiting their mitochondria function. In addition, Q/D effectively reduced clonogenicity, blocked cell-cycle progression, and promoted apoptosis [122]. Understanding the interplay between mitochondria and cancer stem cells will provide better clues to what may constitute new treatment strategies. One possible explanation is through removing the mitochondria’s support from HIF-1 action. Emerging evidence has shown the central role of HIF-1 in CSCs generation and maintenance (see Section 2.1) [26,50]. A proposition of this concept is that targeting cancer-specific mitochondria and removing their support from HIF molecules can help to eliminate CSCs and decrease the rates of tumor recurrence and metastasis.

### 3.5. Future Directions

Thanks to the current understanding of mitochondria’s role in cancer metabolism, anti-mitochondrial therapy can be a potential therapeutic approach in oncology. It can serve as an adjuvant to radiotherapy by preventing ROS clearing, adjuvant to chemotherapy by inactivating cell-cycle arrest, efflux pump, and ROS clearing, and adjuvant to ITx by preventing mitochondria hijacking (see Section 2.4). Anti-mitochondria therapy has the potential to serve as a definitive therapy as well. This can be mediated by inhibiting the pathways that are the cornerstone of cancer-cell metabolism to survive and progress. By completely inhibiting mitochondrial function, at least twenty-two vital mechanisms become synchronously affected (Figure 1). There is possibly no way for cancer cells to circumvent the anti-mitochondrial therapy because there is no similar organelle to respond to their requirements. Therefore, resistance to anti-mitochondrial therapies is hard for cancer cells. Moreover, anti-mitochondrial therapy can remove/weaken the support from the surrounding cells, including MDSCs and Tregs (see Section 2.2.4), which improve cancer cells’ survival, progression, and resistance. In this condition, the cancer cell cannot survive in the hypoxic and acidic TME, evade the immune system, and improve its malignancy. Therefore, anti-mitochondrial therapy can revolutionize future cancer treatment.

Accumulating evidence indicates that cancer cells can maintain the mitochondria ultrastructure and function in hypoxic conditions [18]. In addition, cancer cells can provide more functional mitochondria for themselves by hijacking from normal cells [2]. By identifying and blocking the mitochondria-boosting pathways, the cancer dilemma is solvable in the future.

## 4. Conclusions

This theory highlighted the importance of mitochondrion in cancer-cell metabolism. It provides crucial benefits for cancer cells in terms of survival in hypoxic TME, immune evasion, progression, and resistance to treatment. Furthermore, cancer cells can maintain their mitochondrial function under hypoxia and even hijack functional mitochondria from normal cells. This paper noted that mitochondrion is the interconnecting ring of different cancer features, such as EMT, stemness, metastasis, drug resistance, radioresistance, and immune evasion. Mitochondria are also involved in the basic metabolism of cancer cells, such as glycolytic flux, protective cell cycle arrest (dormancy), autophagy, and quiescence evasion. With these things in mind, mitochondria are necessary for cancer cells to survive. Given their multifaceted role in cancer cells, mitochondria are possibly *cancer’s Achilles’ heel*. Practitioners can overcome cancer by identifying and blocking the strategies by which cancer cells maintain their mitochondria’s quality and quantity. Further studies are warranted to examine this theory.

## Figures and Tables

**Figure 1 genes-13-01728-f001:**
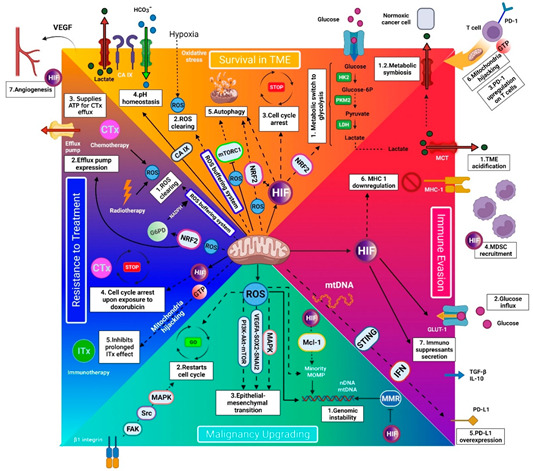
Schematic model of mitochondria’s role in cancer survival, immune evasion, progression, and treatment resistance. The white boxes depict the mitochondria regulation outcomes. (A) Survival in the tumor microenvironment (TME) (orange area): Functional mitochondria are requisite for cancer cells to survive in the harsh TME by facilitating/mediating (A1) glycolysis, (A2) ROS clearing, (A3) cell cycle arrest, (A4) enhanced pH homeostasis, (A5) autophagy, (A6) mitochondrial hijacking, and (A7) angiogenesis. (B) Immune evasion (purple area): Mitochondria assist cancer cells in evading the immune cells by mediating (B1) TME acidification, (B2) glucose influx, (B3) PD-1 upregulation on T cells (by mitochondrial hijacking), (B4) recruiting myeloid-derived suppressor cells (MDSCs), (B5) PD-L1 overexpression on cancer cells, (B6) MHC-1 downregulation, and (B7) immunosuppressants secretion. Moreover, mitochondrial hijacking from T cells depletes T cells’ energy and impedes long-term activity against cancer. (3) Malignancy upgrading (light blue area): Mitochondria are essential for cancer progression by mediating (C1) genomic instability, (C2) quiescence evasion, and (C3) epithelial-to-mesenchymal transition. These actions are mediated by reactive oxygen species (ROS) production. (D) Resistance to treatment (dark blue area): (D1) Mitochondria can serve as a defense shield for cancer cells against radiotherapy and chemotherapy by clearing ROSs. (D2–4) Moreover, they improve chemotherapy resistance by mediating efflux pump expression, providing ATP for efflux pumps, and inducing cell cycle arrest. (D5) In addition, mitochondria hijacking from T cells impairs the long-term effect of anti-PD-1 immunotherapy. Note: The HIF- and GTP-mediated extracellular outcomes are shown in their corresponding white boxes. ATP indicates adenosine triphosphate; CA IX, carbonic anhydrase IX; EMT, epithelial–mesenchymal transition; FAK/Src/MAPK, focal adhesion kinase/Src/mitogen-activated protein kinase; GLUT-1, Glucose transporter-1; GTP, guanosine triphosphate; G6PD, glucose 6-phosphate dehydrogenase; HIF, hypoxia-inducible factor; HK2, hexokinase 2; IFN, interferon; IL-10, interleukin-10; LDH, lactate dehydrogenase; MDSC, myeloid-derived suppressor cell; MHC-1, major histocompatibility complex class I; mTORC1, mechanistic target of rapamycin complex 1; mtDNA, mitochondrial DNA; NADPH, nicotinamide adenine dinucleotide phosphate; NRF2, nuclear factor-erythroid 2 related factor 2; PI3K/Akt/mTOR, phosphatidylinositol-3- kinase/protein kinase B/mammalian target of rapamycin; PD-1, programmed cell death protein-1; PD-L1, programmed cell death protein-ligand 1; PKM2, pyruvate kinase M2; ROS, reactive oxygen species; STING, stimulator of interferon genes; TGF-β,transforming growth factor-beta; TME, tumor microenvironment; VEGF, vascular endothelial growth factor; VEGFA/SOX2/SNAI2, vascular endothelial growth factor A-SRY-Box Transcription Factor 2.

## Data Availability

Not applicable.

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
