# Peer review of "Targeted Anti-Mitochondrial Therapy: The Future of Oncology"

_genes, 2022, doi:10.3390/genes13101728_

Round 1

Reviewer 1 Report

Aerobic glycolysis is an important metabolic adaptation of cancer cells. However, there is growing evidence that reprogrammed mitochondria also play an important metabolic and regulatory roles in sustaining and promoting tumor progression.

In the current review the authors decided to focus on this very important observation and to  thoroughly and thoughtfully discuss the growing validity of mitochondria as a promising target for cancer therapy. The authors systematically discuss the various mitochondrial roles enabling the survival of growing tumors in metabolically  challenged  tumor-microenvironment  (TME), which also  serves as wrestling arena between the tumor cells and the immune surveillance and targets killing, by T and NK, cells. Furthermore, the authors describe and discuss the roles of mitochondria in supporting therapy evasion and resistance acquirement by cancer cells. Overall, this is an educating and important review which puts the focus on a rising potential target for cancer therapy. However, it seems that the authors focus their literature scan and discussion, primarily on solid tumors. Thus, it will be beneficial if the authors would devote at least one additional section to the roles played by  mitochondria in non-solid tumors, i.e. Leukemias etc. This would turn this review rather comprehensive and widely presenting this important topic.  

Author Response

Responses to the Esteemed Reviewer 1’s comments:

General overview: Aerobic glycolysis is an important metabolic adaptation of cancer cells. However, there is growing evidence that reprogrammed mitochondria also play an important metabolic and regulatory role in sustaining and promoting tumor progression.

In the current review the authors decided to focus on this very important observation and to thoroughly and thoughtfully discuss the growing validity of mitochondria as a promising target for cancer therapy. The authors systematically discuss the various mitochondrial roles enabling the survival of growing tumors in metabolically challenged tumor-microenvironment (TME), which also serves as wrestling arena between the tumor cells and the immune surveillance and targets killing, by T and NK, cells. Furthermore, the authors describe and discuss the roles of mitochondria in supporting therapy evasion and resistance acquirement by cancer cells. Overall, this is an educating and important review which puts the focus on a rising potential target for cancer therapy.

Response: We sincerely thank you for your effort and time in evaluating and commenting on our manuscript. We are delighted about the constructive feedback and appreciate your helpful advice. After profoundly editing the manuscript according to the reviewers’ remarks, we are convinced that the quality of the manuscript has significantly improved and hope that the revision is now suitable for publication in Genes. In the revised manuscript, we added 3740 words and 41 references to respond better to the reviewers’ comments. Below text is the point-by-point response to your valuable comment. Besides, the changes are addressed and can be tracked in the revised manuscript.  

Comment 1: However, it seems that the authors focus their literature scan and discussion, primarily on solid tumors. Thus, it will be beneficial if the authors would devote at least one additional section to the roles played by mitochondria in non-solid tumors, i.e. Leukemias etc. This would turn this review rather comprehensive and widely presenting this important topic. 

Response: We appreciate this insightful comment. It helped us to improve our manuscript. To maintain the contents’ coherency, we added the relevant studies regarding the leukemia/lymphoma under the corresponding subheadings., as follows:

2.1.1. Metabolic switch to glycolysis: (lines 100-104)

In a study on JAK2V617F-positive myeloproliferative neoplasms, HIF-1 signaling was involved in the regulation of genes promoting glycolysis (glucose transporters 1 and 3 [GLUT-1 and -3], phosphofructokinase/fructose-bisphosphatase 3 [PFKFB3], and lactate dehydrogenase A [LDHA]) and shunting pyruvate from TCA cycle (PDK1).

2.1.2. Redox homeostasis: (lines 138-144)

A study on chronic lymphocytic leukemia (CLL) cells demonstrated another interaction between intracellular ROS and mitochondria. This study indicated that oxidative stress in CLL cells induced the overexpression of stress-responsive heme-oxygenase-1, which in turn promoted mitochondria biogenesis. In other words, oxidative stress, ROS buffering system, and mitochondria form a virtuous circle in CLLs, promoting cancer pathogenesis in different pathways (see sections 2.3 and 3.3).

2.1.3. Protective cell cycle arrest (dormancy or quiescence): (lines 164-169)

In leukemic stem cells, quiescence is regulated by the mTOR signaling pathway. Evidence has shown that moderate ROS levels (denoting mitochondrial activity) promotes the mTOR pathway. It has been demonstrated that a competent redox system in cancer cells maintains the ROS balance higher than normal cells at a moderate level to pro-mote cancer progression (see section 2.3). Therefore, mitochondria can participate in the dormancy process via distinct signaling pathways.

2.1.5. Autophagy: (line 199)

A similar mechanistic pathway (Akt/mTORC1) regulates autophagy in leukemic cells. So, in the revised manuscript the following reference is added:

  1. Rothe, K.; Porter, V.; Jiang, X. Current Outlook on Autophagy in Human Leukemia: Foe in Cancer Stem Cells and Drug Resistance, Friend in New Therapeutic Interventions. Int J Mol Sci 2019, 20, doi:10.3390/ijms20030461.

2.1.7. Angiogenesis: (lines 252-254)

In a study on EL4 lymphoma cells, a combination of angiostatin and anti-HIF agents was more effective in restricting the lymphoma cells’ growth compared with angiostatin alone.

2.2.1. TME acidification: (lines 285-287)

“In a study on acute lymphoblastic cells, Bohloli et al. demonstrated that acidic TME helped leukemic cells proliferation, invasion, and resistance to apoptosis by doxorubicin (an anthracycline agent).”

2.2.2. Glucose influx: (lines 295-296)

Also, in JAK2V617F-positive myeloproliferative neoplasms, HIF-1α is the primary regulator of GLUT-1 and -3 expression.

We genuinely appreciate your positive feedback and valuable comment to help to improve our manuscript. We tried our best to address the comment and hope that the corrections could meet the requirements for approval.

Sincerely yours

The corresponding authors

Reviewer 2 Report

The manuscript “Targeted Anti-mitochondrial Therapy: The Future of Oncology” by Taghizadeh-Hesary F. et al focus the attention to the several functions and steps in which mitochondria are involved in cellular processes and that may be valuable targets or fuels to explore in antitumorigenic research.

Overall, the manuscript is well written and the structure is well thought by the authors. Still, I have some comments and suggestions that I list below.

Major:

1. As a “Commentary: opinion & Hypothesis” paper, the paper needs to be more critical on the literature reports that are described and less based on description. The authors mostly provide description on the different topics and subtopics without commenting why (or why not) is that important in cancer and in antitumoral strategies along the entire manuscript. 

Besides that, the authors should (based on their comments on the literature reports) suggest some potential ways how to use or avoid each kind of mechanism/pathway/role thar could benefit patients (as hypothesis).

2. Some sections are really reduced and based in 2/3 main articles. Although some are very new (and probably there are few reports in these subjects; as should be the case of mitoc hijacking), but others there are much more information that should be described:

2.1.3. Protective cell cycle arrest (dormancy or quiescence)

2.1.4. pH homeostasis

2.1.6. Mitochondria hijacking

2.1.7. Angiogenesis

2.2.1. TME acidification

2.2.2. Glucose influx

2.2.6. Defective antigen presentation

2.2.7. Immunosuppressive mediators

2.4.2. Radiotherapy

3. The authors discuss the involvement of ROS in several processes. Nevertheless, I believe that the authors forgotten to discuss that ROS are considered by some as secondary signaling molecules. 

I suggest the authors to add a new section discussing this subject and, if the authors believe it is important, in how it might also be valuable in cancer (or not).

Minor comments:

1. “A recent in vitro experiment from the United States showed that cancer cells are dependent on normal cells for their living and function”

1.1. Is it really important to mention the country where the study was conducted? If yes, please state why! 

1.2. There is no reference for such study in this phrase. Please add!

2. In the next phrase, the authors state that cancer cells can receive mitochondria from immune cells. Perfect. But what about other “normal cells”? What about tumor associated fibroblasts or cells from cancer microenvironment? Please add some more info regarding other “normal cells” from whose tumor cells are also dependent for their living and function.

3. “An in vitro study on 108 hepatocellular carcinoma cells demonstrated that cancer cells could cope with this condition by removing the accumulated ROS”.

This sentence is too vague bur refers to a study. I recommend one of two possibilities.

3.1. Or the authors add more detaisl on how the hepatocellular cancer cells cope with this, i.e. how do they remove ROS, OR

3.2. The authors state a general sentence and refer this study as one example how cancer cells overpass the ROS accumulation.

4. “Besides genetic mutations, the in-257 activation of DNA damage repair pathways is essential to establish the genomic instability 258 in cancer cells.” Please add reference!

Author Response

Responses to the Esteemed Reviewer 2’s comments:

General overview: The manuscript “Targeted Anti-mitochondrial Therapy: The Future of Oncology” by Taghizadeh-Hesary F. et al focus the attention to the several functions and steps in which mitochondria are involved in cellular processes and that may be valuable targets or fuels to explore in antitumorigenic research. Overall, the manuscript is well written and the structure is well thought by the authors. Still, I have some comments and suggestions that I list below.

Response: Response: We sincerely thank you for your effort and time in evaluating and commenting on our manuscript. We are delighted about the constructive feedback and appreciate your helpful advice. After profoundly editing the manuscript according to the reviewers’ remarks, we are convinced that the quality of the manuscript has significantly improved and hope that the revision is now suitable for publication in Genes. In the revised manuscript, we added 3740 words and 41 references to respond better to the reviewers’ comments. Below text is the point-by-point response to your valuable comment. Besides, the changes are addressed and can be tracked in the revised manuscript.  

Major comments:

Comment 1: As a “Commentary: opinion & Hypothesis” paper, the paper needs to be more critical on the literature reports that are described and less based on description. The authors mostly provide description on the different topics and subtopics without commenting why (or why not) is that important in cancer and in antitumoral strategies along the entire manuscript. Besides that, the authors should (based on their comments on the literature reports) suggest some potential ways how to use or avoid each kind of mechanism/pathway/role that could benefit patients (as hypothesis).

Response to major comment 1: We appreciate this insightful comment. It helped us to improve our manuscript. In response, we added a paragraph to each subheading denoting the interpretation and clinical implications. In addition, we run more comprehensive literature review to add recently-published, relevant studies:

2.1.1. Metabolic switch to glycolysis: (lines 114-119)

This section denoted how functional mitochondria can support HIF-1α stability and function to improve the cancer cells’ capacity to run glycolysis, responding to their high energy demands, proliferation, and surviving in the hypoxic TME. Therefore, targeting the cancer cells’ mitochondria can likely reduce their capacity to proliferate and survive in the harsh TME, and makes the overall prognosis better.

2.1.2. Redox homeostasis: (lines 148-152)

Therefore, targeting cancer-specific mitochondria can potentially reduce the cancer cells’ capacity to survive in the oxidative TME or to progress (CLL). Besides, evidence has shown that the maintenance of ROS homeostasis determines the maintenance of the CSCs’ phenotype. Hence, tumor recurrence and metastasis rates will likely reduce by targeting the cancer-specific mitochondria.

2.1.3. Protective cell cycle arrest (dormancy or quiescence): (lines 173-175)

Thereby, inhibiting the cancer cells’ mitochondria would likely increase the rates of proliferative cancer cells, which in turn improves radiotherapy and chemotherapy efficacy.

2.1.4. pH homeostasis: (lines 190-192)

Therefore, targeting cancer cells’ mitochondria can disable cancer cells to provide pH homeostasis surviving in an acidic TME.

2.1.5.   Autophagy: (lines 204-207)

By inhibiting mTORC1 and activating HIF-1α and NRF2, mitochondria are likely a master regulator of autophagy in cancer cells. Thereby, inhibiting mitochondria would likely impede this rejuvenation strategy of cancer cells and sensitize them to the imposing stresses.

2.1.6.   Intercellular mitochondria trafficking: (lines 218-2221 and 233-237)

Targeting mitochondria can likely block the mitochondria’ migration through the nano-tubes; thereby shifting the energy balance toward the tumor-fighting immune cells (see section 3.1), which can improve the response to anti-programmed cell death protein-1 (PD-1) agents.

Overall, functional mitochondria can also participate in mitochondria transferring be-tween cancer cells as well. Therefore, targeted anti-mitochondrial therapy would likely interfere with mitochondria trafficking between cancer cells, reducing their pathogenesis. Further studies can reveal this notion.

2.1.7.   Angiogenesis: (lines 251-256)

Therefore, anti-angiogenic agents require complementary agents to impede HIF-1 expression or activity. In a study on EL4 lymphoma cells, a combination of angiostatin and anti-HIF agents was more effective in restricting the lymphoma cells’ growth compared with angiostatin alone. Another potential choice for this aim is anti-mitochondrial therapies that remove the mitochondrial support from the HIF activity (see section 2.1.1).

Section 2.1: (lines 258-265)

Collectively, this section demonstrated that functional mitochondria are vital for cancer cells to survive in a harsh TME. Notably, this section also addressed that CSCs have great capacity to run glycolysis and buffering ROSs. It has been evidenced that HIF-1α is the dominant regulator in generation and maintenance of CSCs. A deep di-ve into the involved molecular mechanisms indicates that HIF-1α upregulates the expression of stem cell factors (e.g., NANOG, OCT4, SOX2, and Krüppel-like factor 4) and TERT gene, an enzyme required for the maintenance of telomeres. Targeting cancer-specific mitochondria and removing their support from HIF molecules can help to eliminate CSCs.

2.2.1. TME acidification: (lines 282-288)

Hence, targeting cancer cells’ mitochondria can unable cancer cells to acidify the TME and helps cancer-specific immune cells recruit the TME, which potentially improves the immunotherapy (ITx) efficacy (see section 2.4.3). In addition, more acidic TME can improve cancer cells’ progression and resistance. In a study on acute lymphoblastic cells, Bohloli et al. demonstrated that acidic TME helped leukemic cells proliferation, invasion, and resistance to apoptosis by doxorubicin (an anthracycline agent). Therefore, mitochondria-targeting agents can improve the chemotherapy efficacy.

2.2.2. Glucose influx: (lines 302-306)

A corollary of this concept is that targeting cancer cells’ mitochondria can reduce glucose uptake by cancer cells, shifting more glucose molecules to the infiltrating immune cells to overcome cancer cells. Besides, limited available glucose for cancer cells can impede the glycolysis flux, essential for their survival, proliferation, and resistance (see sections 2.1.1 and 2.1.2).

2.2.3. Mitochondrial hijacking: (lines 314-321)

This understanding addresses a crucial concept that blocking mitochondria biogenesis can disable the mitochondria hijacking process. This action can improve anti-tumor immunity in two ways: (1) by shifting the energy balance toward the immune cells to over-come cancer and (2) by reducing the expression of PD-1 molecules on T cells. Besides, mitochondria hijacking can reduce the immune cells’ ability to provide prolonged interaction with cancer cells in the context of ITx. Therefore, the improvement of ITx is fore-seeable with targeted anti-mitochondria therapy and blocking the hijacking process.

2.2.4. Recruitment of myeloid-derived suppressor cells (MDSC) toward TME: (lines 333-340)

Therefore, one may conclude that targeting cancer cells’ mitochondria can impede the recruitment of MDSCs into TME, which in turn reduces the number of Tregs in TME. In addition, there is evidence that cancer cells support the Tregs function and proliferation by providing lactate. Sections 2.1.1 and 2.1.4 noted that lactate production (by glycolysis) is HIF-dependent and lactate efflux (by MCT molecules) is CA IX-dependent. Given that HIF and CA IX molecules require mitochondria support for the proper action, blocking mitochondria can potentially impede the Tregs activity and proliferation in TME.

2.2.5. Expression of immune checkpoints: (lines 353-357)

In summary, cancer cells’ mitochondria participate in PD-L1 expression on cancer cells and PD-1 expression on T cells, while MDSCs’ mitochondria participate in PD-L1 expression on MDSCs. This finding indicates that targeting MDSCs’ mitochondria can also help to improve the immune response by reducing PDL1-induced immune inactivation.

2.2.6. Defective antigen presentation: (lines 372-375)

Hence, targeting cancer cells’ mitochondria removes their support from HIF-1α and MAPK and enhances the antigen presentation to T cells. In this condition, more immune cells infiltrate the TME. More immune cell infiltration would enhance both intrinsic (without ITx) and ITx-induced immune responses.

2.2.7. Immunosuppressive mediators: (lines 385-389)

These findings note that mitochondria can mediate the production of immunosuppressants through multiple pathways. Therefore, mitochondria blocking would turn the immunosuppressive TME into proinflammatory gene signature. This change can improve the response to ITx and radiotherapy, both through enhancing effector T cell infiltration into TME.

Section 2.2: (lines 391-393)

Collectively, this section demonstrated that functional mitochondria are crucial for cancer immune evasion. Targeting cancer cells- and MDSCs-specific mitochondria can improve the anti-tumor immunity in the context of ITx and intrinsic immunity.

2.3.1. Genomic instability: (lines 421-424)

This paragraph illustrated how functional mitochondria can induce and preserve genetic mutations in cancer cells. With a gradual increase in genomic instability, cancer cells lost their differentiation. This process will be translated to high proliferation and invasion in cancer behavior.

2.3.2. Quiescence evasion: (lines 445-448)

Looking back to section 2.1.3, we may conclude that mitochondria are involved in both cell cycle arrest and cell cycle evasion. Further studies are required to reveal underlying mechanistic pathways involved in this dual behavior of mitochondria in different TME conditions.

*** In this paragraph, we also updated the text to follow the recent studies, as follows: (lines 428-437)

This process is similar to aerobic glycolysis, in which HIF-1α shifts cancer cells’ metabolism from OXPHOS to glycolysis. However, under normoxic conditions, HIF-1 is deactivated. Therefore, one may conclude that quiescence evasion is orchestrated by an-other transcription factor. Recent evidence on a non-small cell lung cancer (NSCLC) model demonstrated that this process is regulated by Ras, Rap1, PI3K/Akt, and ERK signaling pathway. Besides, it has been noted that mitochondria are indexed actors in Ras-mediated cancer proliferation. Serasinghe et al. indicated that mitochondria dynamics, shifting from mitochondrial fusion into fission, is required for cancer proliferation. Therefore, determining factors involved in mitochondria dynamics can help to over-come Ras-mediated quiescence evasion.

2.3.3. Metastasis: (lines 465-466)

Therefore, targeted anti-mitochondrial therapy has the potential to disrupt EMT and metastasis.

Section 2.3: (lines 468-471)

Collectively, this section demonstrated how functional mitochondria assist cancer progression. Targeting cancer-specific mitochondria can reduce their ability to de-differentiate, proliferate, and metastasize. Therefore, it can help to improve the treatment results and overall prognosis.

2.4.1. Chemotherapy: (lines 490-493)

Therefore, functional mitochondria enhance the cancer cells’ chemoresistance through (at least) five mechanisms. Future clinical trials can examine anti-mitochondrial therapies to improve chemotherapy efficacy.

*** In this paragraph, we also added a new mechanism of mitochondria for chemoresistance, as follows: (lines 489-490)

(5) and, as alluded to above (section 2.2.1), mitochondria can help to doxorubicin-resistance. by mediating TME acidification.

2.4.2. Radiotherapy: (lines 503-509)

A corollary of these concepts is that anti-mitochondrial therapy can serve as a radiosensitizer by reducing the ROS-clearing capacity of cancer cells and improving effector T cell infiltration. Therefore, targeted anti-mitochondrial therapy can serve as a radiosensitizing agent. The clinical implication of this concept is to reduce the prescribed radiation doses, which reduces the treatment toxicities and improves the quality of life. Future preclinical and clinical studies can reveal the radiosensitizing efficacy of anti-mitochondrial therapies.

2.4.3. Immunotherapy: (lines 518-521)

Therefore, targeted anti-mitochondrial therapy can improve the immune cells’ energy content to better recruit the TME and counteract the cancer cells. This strategy can play as an adjuvant to diverse ITx approaches.

Section 2.4: (lines 523-525)

Collectively, this section demonstrated how anti-mitochondrial therapies can potentially tackle cancer treatment resistance.

3.1. An Energy Battle Between Immune and Cancer Cells: (lines 554-555)

This theory might improve the ITx efficacy in future studies.

3.2. Mitochondria Improve Treatment Resistance: (lines 579-582)

This article indicated that anti-mitochondrial therapy can block/attenuate multiple resistant mechanisms to improve the treatment responses for the available anti-cancer approaches, including chemotherapy, radiotherapy, ITx, and targeted therapies.

3.3. Reactive-oxygen Species: The Main Weapon of Mitochondria: (lines 597-599)

Given the diverse and crucial effects of mtROS as signaling molecules, anti-mitochondrial therapies serve as a potential anti-cancer treatment influencing cancer survival, progression, and resistance. 

3.4. Cancer Stem cells Can Be Defeated by Targeting Mitochondria: (lines 613-615)

A corollary of this concept is that targeting cancer-specific mitochondria and removing their support from HIF molecules can help to eliminate CSCs and decrease the rates of tumor recurrence and metastasis.

*** In addition, in this section we added the mechanism how mitochondria are involved in CSCs biology, as follows: (lines 611-613)

One possible explanation is through removing the mitochondria’s support from HIF-1 action. Emerging evidence has shown the central role of HIF-1 in CSCs generation and maintenance (see section 2.1).

2.4. Future Directions

In this section, we highlighted the importance of tumor microenvironment components in cancer biology and resistance, and noted how anti-mitochondrial therapy can disrupt their support for cancer progression, as follows: (lines 625-631)

It is possibly no way for cancer cells to circumvent the antimitochondrial therapy because there is no similar organelle to respond to their requirements. Therefore, resistance to anti-mitochondrial therapies is hard for cancer cells. Besides, antimitochondrial therapy can remove/weaken the support from the surrounding cells, including MDSCs and Tregs (see section 2.2.4), which improve cancer cells’ survival, progression, and resistance.

Comment 2: Some sections are really reduced and based in 2/3 main articles. Although some are very new (and probably there are few reports in these subjects; as should be the case of mitoc hijacking), but others there are much more information that should be described:

Response to major comment 2: We appreciate this valuable comment. It helped us to provide more evidence supporting our hypothesis. In response to your comment, we added at least two another reference (with contents) to each noted section, as follows:

2.1.3. Protective cell cycle arrest (dormancy or quiescence): (lines 155-157, 159-169)

In addition, dormancy is a strategy for metastatic cells by which they can remain quiescent in tissues for up to twenty years to find a favorable condition for restarting the cell cycles to proliferate.

As noted above, HIF-1 requires mitochondria support for proper activity. Another study on prostate cancer cells indicated an alternative dormancy signaling pathway, the mitogen-activated protein kinases (MAPK)-dependent p38 activation. It has been evidenced that an increase in mtROS (denoting mitochondria biogenesis) can activate the MAPK-p38 pathway. In leukemic stem cells, quiescence is regulated by the mTOR signaling pathway. Evidence has shown that moderate ROS levels (denoting mitochondrial activity) promotes the mTOR pathway. It has been demonstrated that a competent redox system in cancer cells maintains the ROS balance higher than normal cells at a moderate level to promote cancer progression (see section 2.3). Therefore, mitochondria can participate in the dormancy process via distinct signaling pathways

2.1.4. pH homeostasis: (lines 182-186 and 188-189)

CA IX helps cancer cells to have an increased intracellular pH and a decreased extracellular pH compared with normal cells, both of which are beneficial for cancer cells. The in-creased intracellular pH facilitates cell proliferation, invasion, and metastasis, while de-creased extracellular pH inhibits immune cells infiltration.

It has been demonstrated that CA IX inhibitors (aromatic sulfonamides and disulfonamides) can restrict the cell growth of different cancer cell lines.

2.1.6. Intercellular mitochondria trafficking (lines 221-237)

On the other hand, tunneling nanotubes can also mediate mitochondria transfer between cancer cells. In a study on bladder cancer cells, Lu et al. demonstrated that mitochondria transferring through tunneling nanotubes from more invasive (T24) toward less invasive (RT4) cells improved cancer proliferation and invasion. This is strong evidence that functional mitochondria are required for cancer pathogenesis, and cancer cells assist each other to promote their ability to proliferate and invade tissues. In addition, Lu et al. indicated that the formation of mitochondria-transferring nanotubes between bladder cancer cells is regulated by Akt/mTOR signaling pathway. There is evidence that mtROSs (denoting mitochondrial biogenesis) are an upstream activator of the Akt/mTOR pathway in moderate ROS levels. It has been demonstrated that a competent redox system in cancer cells maintains the ROS balance higher than normal cells at a moderate level to promote cancer progression (see section 2.3). Overall, functional mitochondria can also participate in mitochondria transferring between cancer cells as well. Therefore, targeted anti-mitochondrial therapy would likely interfere with mitochondria trafficking between cancer cells, reducing their pathogenesis. Further studies can reveal this notion.

2.1.7. Angiogenesis: (lines 246-256)

Therefore, one may conclude that anti-mitochondria therapy is more effective than anti-VEGF agents (e.g., bevacizumab) due to inhibiting multiple angiogenic factors. Notably, successfully blocking the angiogenesis can inversely promote cancer proliferation and invasion. This counteracting effect may occur secondary to intra-tumoral hypoxia and in-creased HIF-1 expression. Therefore, anti-angiogenic agents require complementary agents to impede HIF-1 expression or activity. In a study on EL4 lymphoma cells, a com-bination of angiostatin and anti-HIF agents was more effective in restricting the lympho-ma cells’ growth compared with angiostatin alone. Another potential choice for this aim is anti-mitochondrial therapies that remove the mitochondrial support from the HIF activity (see section 2.1.1).

2.2.1. TME acidification: (lines 270-274, and 282-288)

Hitherto, it was believed that TME acidification is a cancer cells’ behavior in response to hypoxia. However, a recent study on breast cancer cells revealed that cancer cells can keep acidification even in normoxia. The investigators revealed that low-pH TME modifies the expression of over 3000 genes responsible for tumor invasion, migration, and survival in acidic pH.

Hence, targeting cancer cells’ mitochondria can unable cancer cells to acidify the TME and helps cancer-specific immune cells recruit the TME, which potentially improves the immunotherapy (ITx) efficacy (see section 2.4.3). In addition, more acidic TME can improve cancer cells’ progression and resistance. In a study on acute lymphoblastic cells, Bohloli et al. demonstrated that acidic TME helped leukemic cells proliferation, invasion, and resistance to apoptosis by doxorubicin (an anthracycline agent). Therefore, mitochondria-targeting agents can improve the chemotherapy efficacy.

2.2.2. Glucose influx: (lines 297-306)

In a study on thyroid cancer cells, Heydarzadeh et al. demonstrated that GLUT-1 overexpression can also be regulated by PI3K/Akt pathway. As noted above (see section 2.1.6), moderate levels of mtROSs (denoting mitochondrial biogenesis) can activate PI3K/Akt signaling pathway. Therefore, functional mitochondria can increase GLUT-1 expression on cancer cells through HIF-1 and PI3k/Akt pathways. A corollary of this concept is that targeting cancer cells’ mitochondria can reduce glucose uptake by cancer cells, shifting more glucose molecules to the infiltrating immune cells to overcome cancer cells. Besides, limited available glucose for cancer cells can impede the glycolysis flux, essential for their survival, proliferation, and resistance (see sections 2.1.1 and 2.1.2).

2.2.6. Defective antigen presentation (lines 360-364 and 367-375)

With this strategy, cancer cells hide their tumor-specific antigens from T cells preventing adaptive immune response.  Defective antigen presentation has been reported in 40-90% of cases of distinct malignancies—including melanoma, colorectal, breast, and cervical cancers, usually associated with poor overall prognosis.

Other oncogenic pathways can participate in MHC-1 downregulation. It has been demonstrated that the MAPK pathway can decrease the surface expression of MHC-I through STAT1 and IRF downregulation. Besides, there is evidence that an increase in mtROSs (denoting mitochondrial activity) can enhance MAPK activity. These findings note that mitochondria can participate in MHC-1 downregulation through multiple pathways. Hence, targeting cancer cells’ mitochondria removes their support from HIF-1α and MAPK and enhances the antigen presentation to T cells. In this condition, more immune cells infiltrate the TME. More immune cell infiltration would enhance both intrinsic (without ITx) and ITx-induced immune responses.

2.2.7. Immunosuppressive mediators: (lines 380-389)

However, HIF-1 is not the only regulator of immunosuppressant secretion. It has been evidenced that the PI3K pathway can also mitigate the immune response to cancer by limiting the secretion of proinflammatory cytokines (e.g., IL-12) and increasing the production of anti-inflammatory cytokines (e.g., IL-10). As noted in section 2.1.6, moderate mtROSs level (denoting mitochondrial activity) promotes PI3K signaling pathway. These findings note that mitochondria can mediate the production of immunosuppressants through multiple pathways. Therefore, mitochondria blocking would turn the immunosuppressive TME into proinflammatory gene signature. This change can improve the response to ITx and radiotherapy, both through enhancing effector T cell infiltration into TME.

2.4.2. Radiotherapy: (lines 498-509)

As noted in section 2.3, cancer cells’ mitochondria keep ROS balance higher than normal cells at a moderate level. An Elevated ROS level activates a focal adhesion kinase (FAK) transcription factor. A recent study on a pancreatic cancer mouse model demonstrated that FAK inhibitors can improve the radiosensitivity of tumors through enhanced CD8+ T cell infiltration. In other words, the FAK signaling pathway induces immunosuppressive TME in pancreatic cancer. A corollary of these concepts is that anti-mitochondrial therapy can serve as a radiosensitizer by reducing the ROS-clearing capacity of cancer cells and improving effector T cell infiltration. Therefore, targeted anti-mitochondrial therapy can serve as a radiosensitizing agent. The clinical implication of this concept is to reduce the prescribed radiation doses, which reduces the treatment toxicities and improves the quality of life. Future preclinical and clinical studies can reveal the radiosensitizing efficacy of anti-mitochondrial therapies.

Comment 3: The authors discuss the involvement of ROS in several processes. Nevertheless, I believe that the authors forgotten to discuss that ROS are considered by some as secondary signaling molecules.

I suggest the authors to add a new section discussing this subject and, if the authors believe it is important, in how it might also be valuable in cancer (or not).

Response to major comment 3: We appreciate this insightful comment. It helped us to improve the Discussion section. The following paragraph is added to the Discussion section, under the subheading of “3.3. Reactive-oxygen Species: The Main Weapon of Mitochondria” between lines 585-599.

This paragraph describes how mitochondria-derived ROSs can serve as signaling molecules in cancer cells. Reactive-oxygen species is an umbrella term denoting highly re-active molecules formed from oxygen (O2). The common forms of ROS are hydrogen peroxide (H2O2), superoxide (O2−), and hydroxyl radical (•OH). ROSs are mainly generated during mitochondrial oxidative metabolism in cancer cells and cancer-associated fibroblasts. Evolving evidence has explored the role of mtROSs, as signaling molecules, in cancer pathogenesis. Hydrogen peroxide molecules can promote different oncogenic signaling pathways, including HIF-1α, PI3k/Akt, and MAPK. As noted in the previous section, these pathways are the main regulators of cancer metabolism in all disciplines of survival in TME, immune evasion, progression, and treatment resistance. Superoxide molecules al-so have signaling effects in cancer cells. It improves the stabilization of HIF-1α and upregulates adenosine monophosphate-activated protein kinase (AMPK), as the main regulator of cancer cells’ growth and proliferation. Given the diverse and crucial effects of mtROS as signaling molecules, anti-mitochondrial therapies serve as a potential anti-cancer treatment influencing cancer survival, progression, and resistance.  

Minor comments:

Comment 1: “A recent in vitro experiment from the United States showed that cancer cells are dependent on normal cells for their living and function”

1.1. Is it really important to mention the country where the study was conducted? If yes, please state why!

1.2. There is no reference for such study in this phrase. Please add!

Response to minor comment 1:

1.1. Thanks for your comment. We found the word ‘United States’ unnecessary, so omitted it (line 33)

1.2. In the original manuscript, the corresponding reference for this phrase was the Saha et al. study, in which the investigators found that cancer cells hijack mitochondria from immune cells for their survival and pathogenesis. To better clarify it, we added the reference to the end of this phrase. (line 34)

Comment 2: In the next phrase, the authors state that cancer cells can receive mitochondria from immune cells. Perfect. But what about other “normal cells”? What about tumor associated fibroblasts or cells from cancer microenvironment? Please add some more info regarding other “normal cells” from whose tumor cells are also dependent for their living and function.

Response to minor comment 2: We appreciate this insightful comment. As the respected reviewer best knows, Saha et al.’s study (Nov 2021) demonstrated a new behavior of cancer cells, mitochondria hijacking from normal cells via nanotubes. This study demonstrated that cancer cells hijack mitochondria from CD8+ T cells and NK cells. However, in this study, no other targeted cells (e.g., cancer-associated fibroblasts, MDSCs, etc.) were demonstrated. Given this notion is new, more studies are required to clarify this new behavior of cancer cells. In the revised manuscript, we added the targeted cell types (i.e., CD8+ T cells and NK cells) to the corresponding sentences (line 36). Hitherto, it was demonstrated that cancer cells can do mitochondria transferring between themselves. We added the corresponding content to section 2.1.6. as follows: (lines 223-237)

“In a study on bladder cancer cells, Lu et al. demonstrated that mitochondria transferring through tunneling nanotubes from more invasive (T24) toward less invasive (RT4) cells improved cancer proliferation and invasion. This is strong evidence that functional mitochondria are required for cancer pathogenesis, and cancer cells assist each other to promote their ability to proliferate and invade tissues. In addition, Lu et al. indicated that the formation of mitochondria-transferring nanotubes between bladder cancer cells is regulated by Akt/mTOR signaling pathway. There is evidence that mtROSs (denoting mitochondrial biogenesis) are an upstream activator of the Akt/mTOR pathway in moderate ROS levels. It has been demonstrated that a competent redox system in cancer cells maintains the ROS balance higher than normal cells at a moderate level to promote cancer progression (see section 2.3). Overall, functional mitochondria can also participate in mitochondria transferring between cancer cells as well. Therefore, targeted anti-mitochondrial therapy would likely interfere with mitochondria trafficking between cancer cells, reducing their pathogenesis. Further studies can reveal this notion.”

Comment 3: “An in vitro study on 108 hepatocellular carcinoma cells demonstrated that cancer cells could cope with this condition by removing the accumulated ROS”.

This sentence is too vague bur refers to a study. I recommend one of two possibilities.

3.1. Or the authors add more details on how the hepatocellular cancer cells cope with this, i.e. how do they remove ROS, OR

3.2. The authors state a general sentence and refer this study as one example how cancer cells overpass the ROS accumulation.

Response to minor comment 3: We selected the 2nd choice. With your kind guide, the corresponding sentence is updated to the following to better clarify the difference between normal cells and cancer cells: (lines 122-126)

“Normal cells cannot tolerate hypoxia due to ROS accumulation. The excess intracellular ROS content causes damage to cellular organelles and biomolecules, including DNA, proteins, and lipids. However, cancer cells can tolerate this condition due to an enhanced redox system, which are provided by functional mitochondria.”

Comment 4: “Besides genetic mutations, the in-257 activation of DNA damage repair pathways is essential to establish the genomic instability 258 in cancer cells.” Please add reference!

Response to minor comment 4: Thanks for this comment. We cited ref. 80, as follows: (line 417)

“80. Pećina-Šlaus, N.; Kafka, A.; Salamon, I.; Bukovac, A. Mismatch Repair Pathway, Genome Stability and Cancer. Front Mol Biosci 2020, 7, 122, doi:10.3389/fmolb.2020.00122.”

We genuinely appreciate your positive feedback and valuable comments to help to improve our manuscript. We tried our best to address the comments and hope that the corrections could meet the requirements for approval.

Sincerely yours

The corresponding authors

Round 2

Reviewer 1 Report

The authors have now adequately addressed my comments. 

Author Response

General overview: The authors have now adequately addressed my comments.

Response: We genuinely appreciate your positive feedback and valuable comment to help to improve our manuscript.

Sincerely yours

The corresponding authors

Reviewer 2 Report

Overall, the authors addressed all the concerns. I strongly believe the manuscript was substantially improved. I thank the authors for their effort and I congratulate them.

Although it will not affect my acceptance decision, I make one additional comment for consideration by the authors, regarding my minor comment 2. I completely agree with the authors argument, but my comment was probably not well explained by me and the authors did not understood my point.

The Introduction paragraph has these phrases:

"A recent in vitro experiment showed that cancer cells are dependent on normal cells for their living and function [2]. In November 2021, Saha et al. demonstrated that cancer cells can hijack mitochondria (the cell's energy factories) from immune cells (CD8+ T cells and natural killer [NK] cells) via nanoscale tube-like structures [2]. Besides providing energy, mitochondria are essential organelles for cancer cells’ survival and evolution. In addition, mitochondria have a pivotal role in cancer stem cells (CSCs) biology, promoting its chemo- and radioresistance [3]. "

The first phrase is general and can be applied to other ways how cancer cells may need normal cells. They do not only "need" mitochondria from NK cells, but this sentence can also be related with their need for microenvironment: extracellular matrix, removal of damaging agents, fueling with required "nutrients" for cancel cell growth, and so on... The authors cite reference 2, but in reality, they should mention study 2 as an example of interest to this study, but first sentence should be without reference.

I suggest two ways:

1. I would make something like "Cancer cells are dependent on normal cells for their living and function. Of particular interest, in November 2021, Saha et al. demonstrated..."

OR

2. Add additional phrase in between mentioning other ways how cancer cells can also be dependent on normal cells (some of which I mention) and then call the attention to this particular study.

Author Response

Response: We sincerely thank you for your effort and time in evaluating and commenting on our manuscript. We are delighted about the constructive feedback and appreciate your helpful advice. We selected the 1st choice to revise the sentence, as follows:

“Cancer cells are dependent on normal cells for their living and function. Of particular interest, in November 2021, Saha et al. demonstrated that cancer cells can hijack mitochondria (the cell's energy factories) from immune cells (CD8+ T cells and natural killer [NK] cells) via nanoscale tube-like structures [2].”

We genuinely appreciate your positive feedback and valuable comments to help to improve our manuscript. We tried our best to address the comment and hope that the corrections could meet the requirements for approval.

Sincerely yours

The corresponding authors
